# Should We Be Screening for and Treating Periodontal Disease in Individuals Who Are at Risk of Rheumatoid Arthritis?

**DOI:** 10.3390/healthcare9101326

**Published:** 2021-10-05

**Authors:** Zhain Mustufvi, Stefan Serban, James Chesterman, Kulveer Mankia

**Affiliations:** 1Leeds Teaching Hospitals NHS Trust, National Institute for Health Research, School of Dentistry, University of Leeds, Leeds LS2 9LU, UK; 2School of Dentistry, University of Leeds, Leeds LS2 9LU, UK; s.t.serban@leeds.ac.uk; 3Leeds Dental Institute, Leeds Teaching Hospitals NHS Trust, Worsley Building, Clarendon Way, Leeds LS2 9LU, UK; james.chesterman@nhs.net; 4National Institute for Health Research Biomedical Research Centre, Chapel Allerton Hospital, Leeds Institute of Rheumatic and Musculoskeletal Medicine, University of Leeds, Chapeltown Road, Leeds LS7 4SA, UK; K.s.mankia@leeds.ac.uk

**Keywords:** rheumatoid arthritis, periodontal disease, periodontitis, prevention, screening

## Abstract

There is increasing evidence supporting an association between periodontal disease (PD) and rheumatoid arthritis (RA), both mechanistically and clinically. Trials have shown that treating PD in people with RA may improve RA disease activity. Patients with musculoskeletal symptoms without arthritis, who test positive for cyclic-citrullinated protein antibodies, are at risk of RA (CCP+ at-risk), with seropositivity preceding arthritis onset by months or years. Importantly, there is evidence to suggest that periodontal inflammation may precede joint inflammation in CCP+ at-risk and, therefore, this could be a trigger for RA. There has been increased research interest in RA prevention and the phenotyping of the pre-RA disease phase. This review will examine the merits of identifying individuals who are CCP+ at-risk and performing screening for PD. In addition, we discuss how PD should be treated once identified. Finally, the review will consider future research needed to advance our understanding of this disease association.

## 1. Introduction

Periodontal disease (PD) is an infective and inflammatory disease of the gums and tooth-supporting tissues. Bacteria form complex biofilms that adhere to the tooth surfaces. In healthy individuals, a harmonious symbiosis exists between the bacterial biofilm and the host’s defenses. In PD, there is a dysregulation of the immune and inflammatory processes, resulting in an amplified response to pathogenic bacteria. This results in damage to the tooth-supporting tissues, including the gingiva (gum), periodontal ligament, and alveolar bone. It is therefore the resulting complex interaction between the host immune system, inflammatory reactions, and a genetic predisposition that leads to PD [1]. Porphyromonas gingivalis (*P gingivalis*) is a gram-negative obligate anaerobe that has been labelled a keystone pathogen for PD in the literature [1]. In mouse models, at low colonisation levels *P gingivalis* has been demonstrated to be able to trigger a dysbiosis, which in turn leads to *P gingivalis*-induced periodontitis and bone loss [2]. In addition, *P gingivalis* has a host of virulence factors, one of which is ‘gingipain’, which enables the pathogen to evade leukocytes, as well as reduce their killing capacity [1].

Untreated PD can lead to pain, infection, and eventual tooth loss. It is estimated that the economic burden of PD in Europe is 150 billion Euros [3], with loss of teeth and resulting edentulism forming a major component of this economic burden.

Rheumatoid arthritis (RA) is an autoimmune inflammatory disease of the joints that results in chronic polyarthritis. RA has a prevalence of approximately 1% [4] and is more common in females [5]. Treatment of RA is through disease modifying antirheumatic drugs (DMARDs), and evidence has shown that early intervention leads to improved disease outcomes [6] with less joint destruction [7]. Rheumatoid arthritis can be a chronic, lifelong, and debilitating condition. Patients with RA can lose their independence and suffer significant social [8] and financial repercussions [9]. In the UK, RA is estimated to cost approximately £3600 per year per individual [10].

Considering the growing body of evidence supporting the associations between RA and PD, this paper aims to examine the opportunities and challenges around the development of integrated care provision between medicine and dentistry in individuals who are at risk of RA.

## 2. The Association between Rheumatoid Arthritis and Periodontitis

There is a growing body of evidence showing an interrelationship between RA and PD. Smoking is a common major risk factor for both diseases. Among other risk factors, genetic factors play a role; it has been shown that both PD and RA severity have been associated with the Human Leukocyte Antigen DRB1 alleles [11]. Obesity has also been reported to be significantly associated with both diseases [12].

The prevalence of PD has consistently been shown to be higher in the RA population as compared with matched controls, with a recent meta-analysis reporting an odds ratio of 1.97 (CI 1.68–2.31) [13]. PD is also consistently more severe in individuals with RA compared with matched controls [14,15,16].

Importantly, increased PD has now been demonstrated in the pre-RA phase, with one study demonstrating a higher prevalence of PD in CCP+ at-risk individuals compared with matched controls [17]. In these same individuals, it was shown that *P gingivalis*, the key periodontopathic bacterium, is more abundant at healthy periodontal sites of CCP+ at-risk individuals compared with healthy periodontal sites of matched healthy controls. In another study, first degree relatives (FDRs) of individuals with RA who were seropositive for ACPA were shown to have a high prevalence and severity of periodontitis [18].

One purported mechanism of RA initiation is the induction of RA related autoantibodies at mucosal sites, including the inflamed periodontal surface [19]. It has been shown that in RA there is an increased citrullination of cytoskeletal filaments due to the action of peptidylarginine deiminase (PAD) enzymes [20], which are capable of converting the amino acid arginine into citrulline, which may lead to the production of ACPAs [21]. A recent study demonstrated that in non-RA individuals with PD, there were high levels of IgA ACPA in the GCF but not in the serum, suggesting that these auto-antibodies were induced locally [22]. Furthermore, the study found a higher prevalence of IgA-ACPA in the GCF in healthy control subjects who had PD compared with healthy controls without PD (25% compared with 15%). *P gingivalis* is the only bacteria known to produce a PAD enzyme (PPAD). The modification of arginine residues at the carboxyl-terminus is unique to *P gingivalis*, and a study has shown that PPAD is able to citrullinate c-terminal arginine residues in the proteins fibrinogen and a-enolase, which are two important autoantibody targets implicated in RA [23].

*Aggregatibacter actinomycetemcomitans* is another periodontal pathogen of interest in relation to citrullination at diseased periodontal sites. *A. actinomycetemcomiticans* has been shown to produce a virulence factor, Leukotoxin A, which is able to induce endogenous peptidyl arginine deiminase (PAD) enzyme production in host neutrophils within the gingival crevicular fluid (GCF) [24]. The ability of this periodontal pathogen to induce local citrullination further points to PD as a risk factor in the development of RA.

The association between PD and RA has been further evidenced through several interventional trials investigating the effect of professional mechanical plaque removal (PMPR), which is the mainstay of periodontal treatment, in patients who have both PD and RA [25,26,27,28]. A meta-analysis found that there was a 0.88 score reduction in DAS-28 following periodontal treatment (95% CI −1.38, −0.38) [29]. One recent randomized controlled trial allocated 107 subjects into 4 arms: group 1, 24 RA patients with PD who had PMPR; group 2, 30 healthy individuals with PD who had PMPR; group 3, 23 RA individuals who had no treatment; and group 4, 30 healthy controls. At 45 days of follow up, group 1 had a reduction in DAS-28 of 1.34 (±0.21) (*p* = 0.011) [16].

## 3. Why Should We Identify Individuals at Risk of Rheumatoid Arthritis?

The benefits of preventing or delaying the development of RA progression are manifold and include a reduced burden of disease; reduced time spent on disease modifying anti-rheumatic drugs (DMARD); and reduced financial burdens for the patient, the health system, and the wider economy. In the literature, there is much emphasis on the “therapeutic window of opportunity”, although the precise definition of this is not entirely clear. Patients with undifferentiated arthritis (UA) and indeed individuals with risk factors in the absence of clinical synovitis may fall within this therapeutic window, where clinical and laboratory abnormalities exist in the absence of RA-classifying signs and symptoms. A meta-analysis investigating the effect of early intervention in patients with UA demonstrated a statistically significant delay in RA development, especially when these patients were treated with methotrexate [30]. 

## 4. How Can We Identify Individuals at Risk of Rheumatoid Arthritis?

Patients with early UA are at risk of developing RA. A clinical prediction rule (CPR) was developed to help predict the likelihood of individuals with UA developing RA [31]. The CPR tool allows patients to be scored by taking into account age, sex, distribution of involved joints, early morning stiffness, tender and swollen joint counts, CRP level, and RF and ACPA positivity. A meta-analysis concluded that a score of ≥9 offers the optimal cut-off point to identify UA patients who have a high risk of RA [32]. 

Individuals with MSK symptoms without clinical arthritis but who test positive for antibodies against citrullinated proteins (CCP+) are at an increased risk of developing RA (‘CCP+ at-risk’). Furthermore, anti-CCP antibodies are the most established risk factor in such individuals, with higher levels of anti-CCP associated with higher risk of progression [33]. Elevations in RA-related autoantibodies are present in individual months and even years prior to the onset of RA, suggesting autoimmunity precedes joint inflammation [34].

CCP+ at-risk individuals can be identified by the screening of first-degree relatives (FDRs) of individuals with RA, for whom the heritability of RA is approximately 20% if ACPA negative or up to 50% if ACPA positive [35]. Patients with a high ACPA titre and who are also rheumatoid factor (RF)-positive are at an increased risk of progression to RA [36]. Clinical symptoms such as prolonged early morning stiffness (EMS) duration also confer an increased risk. If present, EMS has been shown to be independently predictive of progression to arthritis [37]. In rheumatology practice, the use of ultrasonography is an important tool used to help risk-stratify at-risk individuals. A positive intra-articular power-doppler signal, detecting early erosions, especially of the feet, has been shown to be predictive of progression to arthritis [38].

## 5. How Should We Screen At-Risk Individuals for Periodontitis?

Periodontal disease screening can be undertaken quickly through a basic periodontal examination (BPE) and can be done by a dentist or a hygienist [39]. Following a positive diagnosis of PD, a more detailed assessment of the gums should be performed, including radiographs, which would determine the level of bone loss around the teeth. 

There are mixed results regarding the evidence for implementing periodontal screening questionnaires. A study comparing questionnaires with full periodontal examinations in a convenience sample of 232 consecutive hospital patients found a diagnostic sensitivity of 78.9% and a specificity of 74.8% [40]. The usefulness of periodontal screening questionnaires has been investigated in two other studies [41,42], citing weaknesses in self-reported periodontal status sensitivity, with one study finding that fewer than half of people with periodontal disease correctly reported having it [43]. A meta-analysis found self-reported PD questionnaires to have acceptable validity, with questions on bleeding gums having a diagnostic OR of 1.4 (CI 0.9–2.2) and for tooth mobility an OR of 11.7 (CI 4.1–33.4) [44].

Questionnaires therefore may be considered useful, though not an alternative to proper periodontal examination. One potential opportunity to utilize questionnaires could be during the patient’s visit to the rheumatologist. A positive screen through the questionnaire would assist with signposting for the dentist or hygienist for further assessment and treatment.

## 6. How Should We Treat PD in Individuals at Risk of RA

The mainstay of periodontal treatment is the mechanical debridement of supra and subgingival plaque and calculus, which form the nidus for periodontopathic infection. Initial treatment is focused on educating the patient regarding relevant risk factors and on the importance of self-performed oral hygiene measures. Oral hygiene instruction is undertaken, including demonstration of effective toothbrushing and interdental cleaning. Professional mechanical plaque removal (PMPR) is then performed, which involves professional debridement of the root surface to remove plaque biofilm and calculus (calcified plaque). This treatment has been shown to be the gold standard for periodontal therapy [45]. Following PMPR, re-evaluation of the disease is essential and is usually undertaken 8–12 weeks later. Depending on the outcome, patients may require repeat treatments or surgical interventions. After a comprehensive periodontal treatment, a life-long commitment to effective self-performed oral hygiene measures and supportive care with PMPR is required. Smoking cessation should also be emphasized, as smoking is a major prognostic factor for PD [46], as well as a risk factor for RA [47]. A meta-analysis of six trials found that the risk of PD progression in quitters was no different when compared with never-smokers (RR 0.97 CI 0.87–1.08) [46]. Conversely, smokers were at an 80% higher risk of PD compared with quitters (RR 1.79, CI 1.36–2.35). 

Trials investigating the effect of PMPR in people with RA have shown it to be effective at reducing PD [25,26], often with positive effects on RA disease activity. To the best of our knowledge, trials investigating the effect of PMPR on individuals at risk of RA are yet to be conducted. 

## 7. Where Should the Periodontal Screening/Treatment Take Place

The screening and treatment of periodontal disease can be undertaken in dental surgery, or alternatively, a bespoke clinic could be considered within the rheumatology department. Due to the simplicity of the screening, it can be undertaken quickly and effectively by the dentist or hygienist in a multi-disciplinary hospital clinic without the need for a dedicated dental surgery [39]. The creation of “integrated practice units” has been advocated to join up care, which requires multiple clinical skills, especially when the care pertains to two coexisting disease conditions [48]. It is argued that this format will defragment care services and reduce the need for multiple patient visits to multiple clinical providers. This form of care has been met with success in a multidisciplinary team setting, including among endocrinologists, hygienists, nutritionists, and physical therapists caring for diabetic patients with periodontitis [39].

## 8. How Should We Measure the Outcome of Treatment?

Following PMPR, it is recommended that the periodontal parameters are repeated after 12 weeks to assess for improvement in PD. The outcome measures for assessing periodontal disease include probing pocket depth, clinical attachment level, bleeding on probing, and plaque scores. 

A number of trials have reported improved DAS28 scores in RA subjects with PD following PMPR. The most appropriate outcome measures for individuals at risk of RA, however, are yet to be determined. Ultimately, the goal is the prevention or delay of arthritis development, but clinical studies and trials may consider using other surrogate outcome measures to demonstrate effects of treatment. Two studies have demonstrated a significant reduction in ACPA titre following PMPR in RA subjects [14,49]; however, two further studies failed to show this [27,50]. Regarding Rheumatoid Factor, several studies have shown that this parameter did not decrease in RA subjects following PMPR [15,27,50,51]. Other biomarkers such as CRP and ESR have been investigated in RA subjects following PMPR. Five studies found no significant reduction in CRP [27,28,52,53,54], and three studies found no significant reduction in ESR [52,53,54]; therefore, these outcome measures are unlikely to be useful. Patient-reported outcomes have been investigated following PMPR, and neither HAQ [28,52] nor early morning stiffness [25,28] were reported to significantly decrease in RA subjects following PMPR. 

Ultimately, delay or prevention of RA would be the key outcome measure for periodontal intervention in CCP+ at-risk patients. Designing an adequately powered trial to investigate this is a challenge, with obstacles including large resource costs and the long study follow-up duration needed to identify patients who progress to RA. Therefore, surrogate outcome measures showing a biological effect of PMPR may need to be used initially. Of the outcome measures, ACPA titre shows the most promise; however, high-quality intervention studies are needed to substantiate this.

## 9. The Acceptability of Identifying and Modifying Risk Factors in Individuals at Risk of RA 

RA is treated primarily through the use of glucocorticoids and conventional synthetic or biological DMARDs. These drugs are associated with many side effects, including infection, gastro-intestinal symptoms, and hepatotoxicity [55]. Methotrexate is commonly regarded as the anchor drug for treating RA. A systematic review of low-dose methotrexate in RA patients found a relative risk of 1.78 (±2.00) in adverse events compared with a relative risk of 1.59 (±1.89) for placebo controls (*p* < 0.001) [56]. 

CCP+ at-risk patients may be reluctant to use pharmacotherapy as a preventative measure. A study interviewing CCP+ at-risk individuals found that many patients are reluctant to use medications, with reasons being worries with drug side-effects and the feeling that medications should be taken only when sick being [57]. An alternative to pharmacotherapy would have obvious advantages, and the answer could be in modifying other risk factors. A study by Sparks et al. demonstrated that, should the risk of developing RA be disclosed to at-risk individuals, they were significantly more likely to modify behaviors, including increased frequency of tooth brushing and flossing [58]. A qualitative study investigating the attitudes of patients with established RA towards periodontal treatment found many barriers, including poor grip strength to hold a toothbrush and poor mobility to attend the dentist [59].

The identification of CCP+ at-risk individuals with PD provides a unique opportunity to modify risk factors, including the shared risk factor of smoking. Smoking cessation has been shown to improve the healing of the periodontium following PMPR and therefore forms an important part of PD management [60]. Smoking cessation has also been shown to be significant in RA. A recent analysis of data from the Nurses’ Health Study in the USA found that with an increasing duration of smoking cessation, there was a decreased risk for RA [61]. 

## 10. A Model for Treating PD in the At-Risk Population 

Multimorbidity, which is the co-existence of two or more chronic diseases, is now well recognized by healthcare professionals, and there is an increased emphasis on treating patients holistically. Dentistry is not always included in these treatment plans [62]. An important parallel to consider is that of the interrelation between periodontitis and diabetes. A trial investigating the effect of periodontal treatment on the glycaemic control of diabetic patients found that following intensive periodontal treatment improved the HbA1C of patients with moderate to severe periodontal disease at 12 months [63]. The trial investigated participants who had moderate-severe PD, and they were randomized into two arms, one of which had intensive periodontal therapy (IPT) and the second of which had routine dental care; cleaning; and polishing above the gumline. The primary outcome measure was the difference in HbA1c at 12 months. After 12 months, HbA1c was 0.6% lower in the IPT group than in the control group (*p* < 0.0001). These data on the effect of periodontal treatment on glycaemic control have led to the early adoption and piloting of a commissioning standard for treating periodontal disease in patients with diabetes in London [64]. In the UK, dental commissioning teams are increasingly exploring innovative approaches to develop targeted services to patients with chronic conditions with the view to bring the “mouth back to the body” [65].

Screening for non-communicable diseases in dental settings is not a new concept. Screening for type 2 diabetes has been investigated in dental clinics. A recent systematic review of nine studies concluded that the use of the dental workforce in screening for type 2 diabetes was beneficial, with a pick-up rate varying between 1.7 and 24%, but called for further high-quality research in this area [66]. It is now appropriate that the acceptability of RA screening in the dental clinic be explored. A qualitative study assessing the beliefs and attitudes of CCP+ at-risk individuals towards having periodontal screening and treatment as a means to improving RA-related outcomes is needed to determine the acceptability of such a service. The results would inform the design of the service and would also help identify potential barriers.

Rheumatologists, dentists, and hygienists should have a heightened awareness of the associations and shared risk factors between RA and PD. Interdisciplinary working should facilitate the holistic management of these patients. A proposed model for screening and treating PD in individuals at risk of RA should not add any additional pressures on resources as it utilizes existing services for patients. The general population are encouraged to have regular dental check-ups, which includes periodontal examination and treatment in general practice. Therefore, the screening and treating of PD in these at-risk groups falls within existing care pathways [67]. Figure 1 shows a proposed pathway for screening and treating PD in individuals at risk of RA.

## 11. Research Agenda

Treating periodontitis in individuals at risk of RA is a novel management option that may contribute to preventing or delaying progression to RA. High-quality intervention trials are now needed to investigate the impact of periodontal treatment in at-risk individuals. In Box 1, the authors make recommendations for future research.

Box 1Recommendations for future research in this area.
**Research Agenda**
Which populations at risk of rheumatoid arthritis should be targeted for periodontal screening and treatment?How should at-risk participants be recruited?Is periodontal screening and treatment acceptable to individuals at risk of RA?What is the threshold of severity of periodontal disease that needs to be treated?What is the acceptable level of periodontal improvement for treatment success?Is periodontal intervention cost-effective for RA prevention?

## 12. Conclusions

RA presents a significant burden, both at the patient and population levels. Measures to decrease the prevalence and severity of RA in the population are therefore much sought-after. In this review paper, we have highlighted that there is considerable evidence of a mechanistic association between RA and periodontitis, particularly in the earliest phase of RA. As a result, we believe that periodontitis should be considered, alongside other predictors, as an important risk factor for RA. Dentists, hygienists, general practitioners, and rheumatologists should have heightened awareness of this when reviewing individuals who are at risk of developing RA. Active screening of PD in at-risk individuals should be considered either in the dental or multidisciplinary setting. We have proposed a model whereby such at-risk individuals can be identified and by which they can receive screening and treatment as appropriate. It is now prudent to implement a trial in individuals at risk of RA to substantiate how efficacious such a treatment model for periodontal disease would be on RA disease progression.

## Figures and Tables

**Figure 1 healthcare-09-01326-f001:**
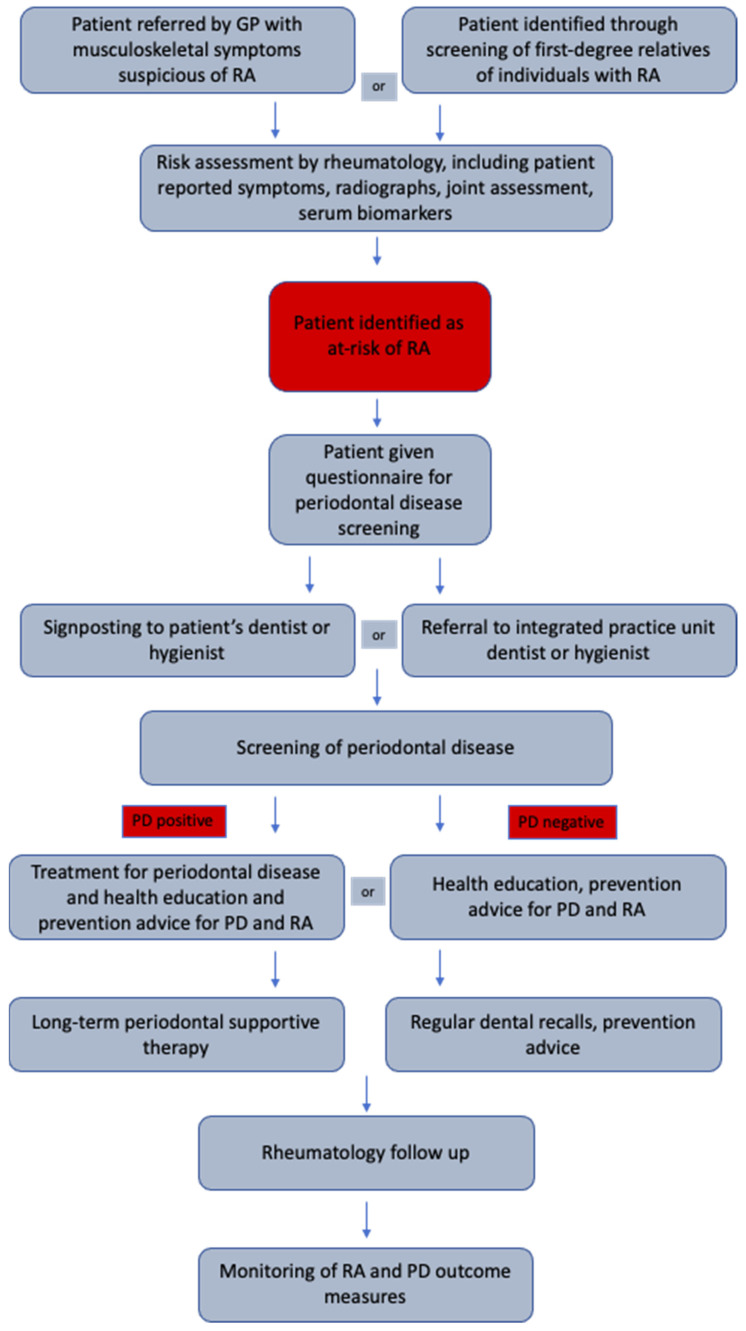
Proposed pathway for screening and treating of periodontal disease for at risk of RA disease patients in secondary care.

## Data Availability

Not applicable.

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
