# Peer review of "Should We Be Screening for and Treating Periodontal Disease in Individuals Who Are at Risk of Rheumatoid Arthritis?"

_healthcare, 2021, doi:10.3390/healthcare9101326_

Round 1

Reviewer 1 Report

On page 2, line 85, line 93 there are the reference numbers and the citations too. Please revise.

On page 2, line 89, P gingivalis in italic. Please revise.

On page 3, line 95, Aggregatibacter actinomycetemcomitans in italic. Please revise, including abbreviation.

On page 4, line 156, is it a question? 

On page 4, line 158, there is just citation, no reference number. Please revise. 

Author Response

Response to Reviewer 1 Comments

Point 1

On page 2, line 85, line 93 there are the reference numbers and the citations too. Please revise.

Response: Thank you for highlighting. This has been corrected.

Point 2

On page 2, line 89, P gingivalis in italic. Please revise.

Response: Thank you for highlighting. This has been corrected.

Point 3

On page 3, line 95, Aggregatibacter actinomycetemcomitans in italic. Please revise, including abbreviation.

Response: Thank you for highlighting. This has been corrected.

Point 4

On page 4, line 156, is it a question? 

Response: Thank you for highlighting. This has been corrected.

Point 5

On page 4, line 158, there is just citation, no reference number. Please revise. 

Response: Thank you for highlighting. This has been corrected.

Reviewer 2 Report

The review has problems with the writing style and is far to verbose. The first paragraph of the introduction is far too long and the emphasis in the writing should match the title. The outline being followed is fine - just too verbose. These styles of paper usually are not fully read by readers due to the long winded. If you can reduce the content to a well directed set of sentences the paper would be reconsidered.

Author Response

Response to Reviewer 2 Comments

Point 1

The review has problems with the writing style and is far to verbose. The first paragraph of the introduction is far too long and the emphasis in the writing should match the title. The outline being followed is fine - just too verbose. These styles of paper usually are not fully read by readers due to the long winded. If you can reduce the content to a well directed set of sentences the paper would be reconsidered.

Response: Thank you for your review. We appreciate the reviewer’s points  and have made the following changes to make the text more concise and focused:

  • Shortened introduction and altered emphasis, including focussing on the question the paper aims to discuss.
  • Shortened section “The association between rheumatoid arthritis and periodontitis”
  • Shortened section on “How should we screen at-risk individuals for periodontitis?”
  • Shortened section on “Where should the screening/treatment take place?”
  • Shortened section on “How should we measure the outcome of treatment?”
  • Shortened section on “The acceptability of identifying and modifying risk factors in individuals at risk of RA”
  • Shortened section on “Research agenda”

We have also made in-text changes to improve the flow of the paper. We hope this meets your approval.

Reviewer 3 Report

I refer to the publication of this review that addressed the importance of identifying individuals at risk for rheumatoid arthritis and screening for PD. In addition, the review proposed further research needed to gain an understanding of this disease association.
The writing is clear, well grounded in the literature. The division into topics makes the article very didactic. The subject is current and with great prospects for advancement.

Author Response

Response to Reviewer 3 Comments

Point 1

I refer to the publication of this review that addressed the importance of identifying individuals at risk for rheumatoid arthritis and screening for PD. In addition, the review proposed further research needed to gain an understanding of this disease association.
The writing is clear, well grounded in the literature. The division into topics makes the article very didactic. The subject is current and with great prospects for advancement.

Response: Thank you for your kind comments. We are pleased you found the paper current and clearly written.